Unravelling approaches to study macrophages: from classical to novel biophysical methodologies

http://orcid.org/0000-0001-8650-8240 Vishnyakova Polina 1 2
http://orcid.org/0000-0002-2392-4439 Elchaninov Andrey 1 2 3
Fatkhudinov Timur 2 3
http://orcid.org/0000-0003-0270-2893 Kolesov Dmitry 1 4 dmitry.v.kolesov@gmail.com
1 National Medical Research Center for Obstetrics, Gynecology and Perinatology Named after Academician V. I. Kulakov of Ministry of Healthcare of Russian Federation , Moscow , Russia
2 Research Institute of Molecular and Cellular Medicine, Peoples’ Friendship University of Russia (RUDN University) , Moscow , Russia
3 A.P. Avtsyn Research Institute of Human Morphology of Federal State Budgetary Scientific Institution “Petrovsky National Research Centre of Surgery” , Moscow , Russia
4 Moscow Polytechnic University , Moscow , Russia
Upadhyay Rohit
Electronic publication date: 2025 Feb 20
Publication date: 2025
Volume: 13
Electronic Location ID: e19039
Received 2024 Oct 3; Accepted 2025 Jan 31
Copyright: © 2025 Vishnyakova et al.
Copyright year: 2025
Copyright holder: Vishnyakova et al.
License: This is an open access article distributed under the terms of the Creative Commons Attribution License, which permits unrestricted use, distribution, reproduction and adaptation in any medium and for any purpose provided that it is properly attributed. For attribution, the original author(s), title, publication source (PeerJ) and either DOI or URL of the article must be cited.
License URL: https://creativecommons.org/licenses/by/4.0/

Keywords: Macrophages, Polarization, Heterogeneity, Methods, Approaches, Techniques, Microfluidics, Nanomechanics

Funding: Moscow Polytechnic University State Assignment “Interorgan Interactions During Liver Regeneration” 123030700110-4 This work was financially supported by Moscow Polytechnic University within the framework of the grant named after Pyotr Kapitsa. The work was supported by the state assignment “Interorgan interactions during liver regeneration” No. 123030700110-4. The funders had no role in study design, data collection and analysis, decision to publish, or preparation of the manuscript.

==============================
Macrophages play crucial roles in immune responses and tissue homeostasis. Despite the fact that macrophages were described more than a century ago, they continue to be the cells of intensive interest. Advanced understanding of phenotypic diversity in macrophages holds great promise for development of cell-based therapeutic strategies. The introduction of innovative approaches in cell biology greatly enhances our ability to investigate the unique characteristics of macrophages. The review considers both classical methods to study macrophages and high-tech approaches, including single-cell sequencing, single-cell mass spectrometry, droplet microfluidics, scanning probe microscopy and atomic force spectroscopy. This review will be valuable both to specialists beginning their study of macrophages and to experienced scientists seeking to deepen their understanding of methods at the intersection of biological and physical sciences.

Introduction

The origin and biology of macrophages

Macrophages constitute a vital component of the immune system, playing multiple key roles in tissue homeostasis and immune defense. Their capacities include phagocytosis, antigen presentation, secretion of soluble factors and interactions with other immune and non-immune cells.

The question of the origin of macrophages, despite almost a century of research, has not been fully worked out and a clear consensus has not been reached. Mammalian macrophages develop from three embryonic sources which correspond to three generations of hematopoietic stem cells (Perdiguero & Geissmann, 2016). As shown in mice the first generation of hematopoietic cells originates in the wall of the yolk sac (Gomez Perdiguero et al., 2015; Perdiguero & Geissmann, 2016) and eventually gives rise to microglia of the central nervous system (CNS) (Hoeffel et al., 2015; Hoeffel & Ginhoux, 2018). The second generation, erythro-myeloid progenitor cells, is thought to descend from hematogenous endothelium of the yolk sac capillaries and give rise to several resident macrophage populations including Kupffer cells of the liver. The third generation of hemopoietic progenitors descend from hemogenic endothelium in the aorto-gonado-mesonephral area and eventually migrate to the liver, bone marrow and other organs, except the CNS.

Macrophages stand out among other mammal cells due to their presence in all organs where they can differ in morphology, function and origin (Park et al., 2022). Thus, alveolar macrophages provide defense against inhaled pathogens and particles; Kupffer cells, specialized liver macrophages, are involved in liver regeneration, blood filtration and bacterial lipopolysaccharide (LPS) clearance; osteoclasts ensure bone resorption; microglia participates in the maintenance of neural homeostasis. There are no macrophages in the so-called barrier compartments, such as the anterior corneal epithelium and the seminiferous epithelium of the convoluted tubules in the testis (Liu et al., 2017; Lokka et al., 2020).

Different populations of macrophages are adapted to specific microenvironments. Macrophages participate in tissue repair by scavenging dead cells and regulating inflammatory responses. With the discovery of the ambivalence of the macrophage phenotype, their role in these processes and also in the development of a particular pathology is rethought. During the development of pathologies such as infections, autoimmune diseases or cancer, macrophages can change their functions and phenotype, which sometimes leads to aggravation of inflammatory processes or contributes to disease progression.

In addition to macrophages directly isolated from experimental animals or patients, macrophage cell lines are often used in studies. RAW264.7 cells are derived from an Abelson leukemia virus-transformed cell line obtained from BALB/c mice (Raschke et al., 1978). This cell line is considered a suitable model for macrophages, RAW264.7 cells are capable of phagocytosis and pinocytosis, and respond to LPS stimulation by releasing various cytokines (Fuentes et al., 2014). Notably, with long-term cultivation, RAW264.7 cells retain their phenotypic and functional stability (including phagocytosis and NO synthesis). However, some authors do not recommend using RAW264.7 cells after 30 passages, due to the pronounced heterogeneity in the studied characteristics (Taciak et al., 2018). Additionally, RAW264.7 cells lack apoptosis-associated speck-like protein containing a caspase activation and recruitment domain, which may interfere with the production of mature IL-1β (Pelegrin, Barroso-Gutierrez & Surprenant, 2008; Zheng, Liwinski & Elinav, 2020; Herb et al., 2024). Another frequently used line of macrophage-like cells is J774.1. This cell line was derived from mouse reticular sarcoma (histiocytic sarcoma) (Ralph & Nakoinz, 1977). J774.1 cells express receptors for immunoglobulins and are capable of phagocytosis and bacterial lysis (Herb et al., 2024).

In addition to macrophage-like cell lines derived from laboratory animals, researchers also use cell lines with properties resembling human macrophages. THP-1 cells were obtained from peripheral blood cells of a 1-year-old boy with acute monocytic leukemia (Tsuchiya et al., 1980). THP-1 monocytes proliferate continuously, leading to the accumulation of mutations and, consequently, heterogeneity (Noronha et al., 2020). Initially, THP-1 was used to study leukemia; however, its ability to differentiate into macrophage-like cells under the influence of phorbol 12-myristate 13-acetate (PMA) or macrophage colony-stimulation factor (M-CSF) has led to its widespread use in macrophage research (Herb et al., 2024). Besides the potential heterogeneity of THP-1-derived macrophages, another characteristic of this line is a lower level of CD14 synthesis compared to human monocytes, which contributes to the low sensitivity of THP-1 cells to LPS (Bosshart & Heinzelmann, 2016). Another cell line derived from human cells is the pro-monocytic line U-937 (Radzun et al., 1983). This line was isolated from a histiocytic lymphoma of a 37-year-old male patient in 1974 (Sundström & Nilsson, 1976). U-937 cells can be differentiated into monocytes or macrophages using various substances, such as PMA, 12-O-tetradecanoylphorbol-13-acetate (TPA), retinoic acids, or 1,25-dihydroxyvitamin D3 (Chanput, Peters & Wichers, 2015).

Understanding the macrophage phenotypic diversity and adaptive responses is a key aspect of modern immunology, opening new horizons for the development of cell-based therapeutic strategies. Expanding the understanding of macrophage phenotypes is impossible without developing new methods for their assessment. At present, laboratories are no longer limited to several markers, as the range of methods for the assessment is expanding. This review highlights new methods for macrophage phenotype assessment at the intersection of cell biology, molecular biology and biophysics.

M1/M2 paradigm, its essence and critique

The fundamental distinction of two polar macrophage phenotypes was introduced by Mills et al. (2000). The paradigm implied the existence of pro- (M1) and anti-inflammatory (M2) macrophages in tissue by analogy with the dichotomy of T helpers 1 and T helpers 2. The use of this nomenclature is common to both human and mouse cells, although not all M1 and M2 markers overlap.

In recent years, the M1/M2 paradigm has increasingly been criticized for its simplification and is beginning to be revised (Blériot, Chakarov & Ginhoux, 2020), which may reflect the development of technologies: high-accuracy measurements indicate that macrophages can spontaneously increase expression of pro- and anti-inflammatory markers even in the absence of stimuli (Specht et al., 2021). Also macrophages could display overlapped phenotypes depending on their tissue location and type of signalling (Strizova et al., 2023).

The strict distinction between M1 and M2 macrophages is gradually being replaced by the idea of continuum or phenotypic spectrum, with M1 and M2 being the extreme and most distinct points, and all intermediate states viewed as transitional phenotypes. Still, the M1 vs. M2 terminology persists as a simple and convenient way to define the vector of macrophage activation. Synonymously, the authors may either indicate the phenotype-inducing stimuli or use the terms ‘pro-’ and ‘anti-inflammatory’ macrophages.

Macrophage polarization depends on many factors including their origin, microenvironmental signals and differentiation commitment. The concept of macrophage phenotype has a biochemical basis related to arginine metabolism. The ornithine cycle, particularly arginase, converts arginine to L-proline required for the collagen synthesis essential in wound healing and polyamines which modulate cell proliferation. Accordingly, high activity of arginase indicates pro-regenerative, anti-inflammatory M2 phenotype. In the opposite case of pro-inflammatory M1 polarization, arginine provides a nitrogen source for the nitric oxide synthesis by NO synthase; the product can either form free nitrogen radicals that act as a bactericidal agent or stimulate soluble guanylate cyclase to generate cyclic GMP within target cells (Rath et al., 2014). It is important to note that up-regulated arginase and NO synthase are markers of polarization in mice, although they are often used by researchers in studies on human cells.

In addition to the arginine fork, the manifestation of the M1/M2 macrophage phenotype is accompanied by a whole complex of molecular changes that can be divided into levels shown in Fig. 1: secreted, surface, and inside the cell molecules. A comprehensive classification of macrophage polarization markers has been proposed by Murray et al. (2014). The methods discussed below are used to evaluate one or several markers at once, which allows researchers to classify a macrophage into one of its polar phenotypes.

Figure 1 Macrophage M1 and M2 markers at different levels within the cell: as secreted, surface, or intracellular molecules in mice, humans, or both.

It is important to note that the manifestation of the phenotype is not limited to proteins alone. The level of non-protein metabolites, reflecting the predominant mode of ATP generation (OXPHOS or glycolysis), phagocytic activity, macrophage motility, and the rigidity of its plasma membrane, are also aspects of the phenotype, for the assessment of which a wider range of methods is used.

Survey methodology

The materials for this review were searched for in the following search engines: Google, PubMed. The search inquiries contained “macrophages” and a method group with consequent refinement of the method subgroup. For example, for microfluidic methods the search inquiries were ‘macrophages microfluidics’ with further refinement ‘macrophages droplet microfluidics’, ‘macrophages organ-on-a-chip’, etc. Reviews of related themes were taken into account. Only articles from the last decade have been reviewed, except classical method-establishing or rare direction works. All selected works were carefully read, and key points were added to the review. Special terms were named following the author’s terminology. In most of the articles, information on cell lines, primary macrophage cultures with their indicated sources of origin, or bacterial strains was provided.

Methods of studying macrophages

To assess the macrophage phenotype, classical methods of molecular and cellular biology are used mainly. In the following, we provide a brief overview of them with a description of the advantages and disadvantages of each technique.

Flow cytometry

Flow cytometry is one of the most common and informative methods for characterizing macrophages. Various protocols and gating strategies have now been developed and applied specifically to characterize primary culture of macrophages from a particular organ, including CNS, lung (Misharin et al., 2013), liver (Daemen, Chan & Schilling, 2021), skin dermis (Forde & Kolter, 2024), spleen, adipose tissue (Silva Ribeiro et al., 2024), etc. Most of the data on the immunophenotype of organ macrophages were obtained on mouse models, and in this regard, the markers of mouse macrophages are presented. It is worth noting that human and mouse macrophages share similar markers, with some exceptions. Regardless of the tissue source the gating strategy in case of mouse macrophages involves pan-leukocyte marker CD45 followed by CD3, CD19 and Ly6G to exclude T cells, B cells and granulocytes, respectively. This is especially relevant for hematopoietic organs such as the red marrow, thymus, spleen, and lymph nodes (Wang et al., 2013). To characterize mice macrophages themselves, a set of markers-CD11b, CD68, CD86, CD163, CD206, F4/80, etc., — is used, which has almost become standard. However, using immunocytochemical markers it is difficult to separate the macrophage population from e.g., the dendritic cells, similar in function and possibly having a close origin. This problem is especially relevant for the population of macrophages and dendritic cells in the kidney (Salei et al., 2020). Therefore, in the mice kidney, these two populations are characterized as cells of a single mononuclear phagocytic system (Kawakami et al., 2013).

Previously, markers such as CD86, CD163, CD206 were used to characterize the functional state of macrophages within the M1/M2 paradigm. However, this approach is currently being abandoned (Murray et al., 2014). To assess the functional state, the intensity of synthesis of certain cytokines is now usually used, which can also be studied using flow cytometry and intracellular staining. TNFα, IL-6, IL-12a (for mouse and human macrophages), IL-1b (for human macrophages) are used as markers for the M1 state and IL-6, IL-10 (only for mouse macrophages) for M2 (Murray et al., 2014).

Specific gene and protein expression analysis

Routine real-time PCR and Western blot techniques with semi-quantitative assessment of gene expression and protein production are one of the most popular methods for rapid detection of up- and down-regulated markers of macrophage activation and polarization (Elchaninov et al., 2021b; Vishnyakova et al., 2021). The advantage of these methods is their speed, low cost, and the possibility of implementation in most laboratories. However, these methods only allow the expression of individual discrete markers to be assessed and do not show the whole picture (Elchaninov et al., 2021a). They are suitable for rapid assessment of macrophage phenotype if considered as a side task in the study. For a more in-depth assessment of processes occurring in macrophages, larger-scale methods are required, such as RNA-sequencing (RNA-seq) of the transcriptome and mass spectrometry of the proteome.

RNA-sequencing

RNA-seq is a common method for transcriptomic analysis. In many cases flow cytometry fails to distinguish cells biologically different but close in origin and functional status, e.g., different macrophage subsets or macrophages vs dendritic cells. In this regard, transcriptomic approaches are more sensitive. In mice RNA-seq allowed, for example, to identify two subpopulations of kidney macrophages and clearly distinguish them from the dendritic cell population (Salei et al., 2020).

RNA-sequencing was particularly useful in studying the origin of resident macrophage populations. Thus, CNS microglia descends from the earliest generation of hematopoietic progenitor cells in the yolk sac wall, whereas resident macrophages in other organs have more diverse sources of origin. Comparative role of developmental sources vs organ microenvironment in the establishment of resident macrophage functionalities is a fundamental issue (Guilliams et al., 2020). The analysis revealed transcriptomic difference between mice macrophages descending from erythro-myeloid precursor cells of the yolk sac wall from macrophages differentiating from blood monocytes (Beattie et al., 2016) despite similar expression levels of mice macrophage markers PU.1, Clec4f and SpiC (Bonnardel et al., 2019).

RNA-seq allows us to assess the expression of a huge number of genes, which greatly expands the capabilities of researchers and our understanding of the functional state of the macrophage population. The disadvantages of RNA-sequencing include high costs and the requirement of specific training for the workflow at all stages of sample preparation and data analysis. Most importantly, the sequencing provides no reflection on the high heterogeneity of macrophages observed both before and during the activation.

Single-cell RNA sequencing

Single-cell RNA sequencing is an approach in which barcoded cDNA libraries are prepared for each individual cell enclosed in a gel bead using microfluidic technology. After sequencing, based on available open annotators or manually by several selected marker genes, distinct subpopulations of cells are identified in samples and whose transcriptome can be analyzed. Widespread single-cell RNA sequencing has become a breakthrough in the study of macrophage phenotype and their heterogeneity populations. The number of published works on this technique deserves a separate review, and we recommend getting acquainted with the already published works (Mulder et al., 2021; Ma, Black & Qian, 2022; Hume, Millard & Pettit, 2023). Using this technique, the heterogeneity and polarization of macrophages in different organs were identified and studied in the following pathologies: in the mice lungs during fibrosis (Aran et al., 2019), in the murine aorta in atherosclerosis (Cochain et al., 2018), in mice acute kidney injury (Yao et al., 2022), in the murine heart in diastolic dysfunction (Panico et al., 2023), in the bronchoalveolar lavage fluid in COVID-19 patients (Liao et al., 2020), in the mice brain in glioma (Ochocka et al., 2021), in a tumor in colorectal cancer patients (Zhang et al., 2020a; Qi et al., 2022) and other types of cancer (Bao et al., 2021; Obradovic et al., 2021; Li et al., 2022b; Xu et al., 2022).

The use of single-cell transcriptomics allows to isolate and analyze macrophages from a sample without cell sorting procedure, to determine their molecular signature and even association with disease prognosis. Articles using this method are published in higher-ranking journals. The disadvantages of the method include high costs, rarity and limited accessibility of the equipment, as well as the complex bioinformatics pipelines.

Mass spectrometry

Along with RNA-sequencing, mass spectrometry provides an advanced tool for studying macrophage heterogeneity, properties and functions (Gudgeon et al., 2024). The use of mass spectrometry has made it possible to distinguish between resident and migrating macrophages that are derived from monocytes. In one study, ten populations of mice organ macrophages (including microglia, Kupffer cells, alveolar macrophages, peritoneal macrophages, spleen red pulp macrophages, intestinal wall macrophages), macrophages derived from red bone marrow monocytes and the RAW264.7 mouse cell line were studied (Qie et al., 2022). Qie et al. (2022) identified 12,205 proteins, including the major transcription factors PU.1, SpiC, etc. A total of 510 transcription factor proteins were identified, including those that provide tissue specificity (Qie et al., 2022). In addition, proteins associated with these transcription factors and involved in the interaction of mice tissue macrophages with the organ environment have been found (Qie et al., 2022). It is interesting to note that resident and migrated macrophages of bone marrow origin in mice differ from each other not only by specific transcription factors, but also by the number of signaling pathways that provide communication of resident macrophages with the tissue environment. Resident macrophages had significantly more such pathways, especially mice alveolar macrophages and Kupffer cells (Qie et al., 2022).

Qie et al. (2022) obtained the correlation of their proteomic analysis with the macrophage RNA sequencing analysis in other works. When comparing the proteome of the studied mice macrophage populations, 40 specific modules containing 35-1,080 proteins were found (Qie et al., 2022). The set of proteins included in one module or another reflects the functional properties of macrophages. Each of the studied mice macrophage populations has its own set of modules, which indicates that organ macrophages are involved in the regulation of functional activity of a particular organ (Qie et al., 2022).

In addition to proteomic analysis, mass spectrometry is used to analyze the lipidome of murine macrophages (Hsieh et al., 2021). Using the RAW264.7 cell line, 400 lipid molecules were identified whose concentration was altered by proinflammatory activation (Dennis et al., 2010).

As already mentioned, proteomic analysis data usually correlate well with RNA-seq. Regulation of protein synthesis at the mRNA level is taken into account. The disadvantages of mass spectrometry are the same as those of RNA-seq: it is time consuming and expensive and requires special equipment, and it is practically impossible to take into account the high heterogeneity of the macrophage population.

Single cell mass spectrometry

The concepts of functional macrophage heterogeneity were expanded using Single-Cell ProtEomics by Mass Spectrometry (SCoPE-MS) (Specht et al., 2021). In the outstanding work of Specht et al. (2021), they analyzed the heterogeneity in a macrophage population that developed from a relatively homogeneous human monocytic precursor, even in the absence of activating cytokines. In this study, 3,042 proteins were analyzed in 1,490 individual monocytes and macrophages over 10 days. A continuous transition of proteomic signatures was detected within the macrophage population (Specht et al., 2021). Next, the SCoPE-MS method was applied to analyze the proteome of LPS-activated murine bone marrow-derived macrophages (Huffman et al., 2023). A total of 1,123 proteins were identified in 373 single primary macrophage cells. Principal component analysis revealed that unexposed macrophages were well distinguishable from macrophages after LPS exposure. However, even after exposure to LPS, macrophages within this group are heterogeneous. Furthermore, it was found that exposure to LPS did not result in marked variability in the content of proteins associated with phagosome maturation, proton transport, and protein targeting to the membrane (Huffman et al., 2023). However, exposure to LPS stimulated variability in proteins related to pathways that regulate the inflammatory response, antigen processing, and presentation via major histocompatibility class (MHC) II and regulation of translational initiation. Huffman et al. (2023) examined how these differences in proteome affect the functional activity of murine macrophages by the uptake of dextran particles. The phagocytosis activity of the dextran particles was found to vary in both groups of macrophages (with and without exposure to LPS) (Huffman et al., 2023). However, the median particle uptake was higher in macrophages treated with LPS. Proteins associated with high levels of dextran particle uptake were identified as mannose receptor C type 1 (MRC1), stabilin 1 (STAB1) and sorting nexin 17 (SNX17). In general, the proteome of murine macrophages with higher dextran particle uptake activity was shown to be more strongly with the proteome of LPS-treated macrophages (Huffman et al., 2023).

Single cell mass spectrometry is a new powerful tool for studying the functional activity of macrophages, and most importantly it takes into account the high heterogeneity of the macrophage population. Among the disadvantages, we can name the difficulty in obtaining samples that meet all the requirements of the method, as well as difficulties in interpreting the obtained data.

FLIM

Fluorescence lifetime imaging microscopy (FLIM) is a powerful imaging technique that can be utilized to assess the metabolic states of macrophages. FLIM can be employed to measure the fluorescence lifetime of endogenous fluorophores, such as NAD(P)H and FAD, which are indicators of cellular metabolism. Regardless of the organism and whether it is a primary culture or cell line M1 macrophages, which are typically associated with a high glycolytic rate and increased production of reactive oxygen species (ROS), may exhibit different fluorescence lifetimes compared to M2 macrophages, which rely more on oxidative phosphorylation and fatty acid oxidation (Szulczewski et al., 2016; Ryabova et al., 2022). By analyzing the fluorescence lifetime of these molecules, researchers can infer the metabolic state of macrophages and correlate it with their functional phenotype. Among the most advanced approaches, we can highlight the two-photon FLIM technique using machine learning for data processing of human blood-derived macrophages (Neto et al., 2022), as well as in vivo visualization of macrophages in human skin (Kröger et al., 2022) and zebrafish (Miskolci et al., 2022). Furthermore, FLIM can be applied in vivo to study macrophage behavior in real-time within their native tissue environments. This metabolic profiling can be crucial for understanding how macrophages respond to various stimuli and their roles in inflammation and tissue repair. Among the advantages of FLIM technique, one can note the vital analysis of samples that does not require their processing and the possibility of functional assessment of cell metabolism even in vivo. The disadvantages of FLIM are the need for expensive equipment, a master method operating on the device, and complex post-processing of images.

The comparison of the methods described is given in Table 1.

Table 1 Comparative characterization of the most common techniques used in macrophage studies.

Method	Opportunities	Strengths	Weakness	References	
Flow cytometry	Proportion of positive cells and the density of antigen both superficial and intracellular	Allows gating of cells to search for a specific population and assess heterogeneity, low cost, widespread, simultaneous staining of several markers in one sample	Need an instrument operator with experience working with macrophage populations, obsolescence of macrophage markers	Wang et al. (2013), Salei et al. (2020), Kawakami et al. (2013), Murray et al. (2014)	
Real-time PCR, Western blot	Semiquantitative assessment of gene/protein expression	Speed, low cost, widespread in laboratories, simple analysis of results	Estimation of the number of genes limited by the capacity of the device and the amount of sample and panel of primers/antibodies	Elchaninov et al. (2021b), Vishnyakova et al. (2021), Elchaninov et al. (2021a)	
Bulk RNA-seq/Mass spectrometry	Transcriptome/proteome analysis of cells or tissue within one sample followed by determination of activated or inhibited signaling pathways after enrichment analysis	Simultaneous detection of multiple genes/proteins per run	High cost, time consuming, difficult equipment availability, need for complex analysis of results	Salei et al. (2020), Guilliams et al. (2020), Beattie et al. (2016), Bonnardel et al. (2019), Gudgeon et al. (2024), Qie et al. (2022), Hsieh et al. (2021)	
Single-cell RNA-seq	Transcriptome analysis of individual cells identified based on their own molecular signature within one sample	Increase the level of publication, a large amount of data from a small number of samples	High cost, time consuming, difficult equipment availability, need for complex analysis of results, need for highly qualified specialists.	Aran et al. (2019), Cochain et al. (2018), Yao et al. (2022), Panico et al. (2023), Liao et al. (2020), Ochocka et al. (2021), Zhang et al. (2020a), Qi et al. (2022), Bao et al. (2021), Obradovic et al. (2021), Li et al. (2022b), Xu et al. (2022)	
FLIM	Fluorescence lifetime of endogenous metabolites after laser excitation	Vital examination of the sample without staining both in vitro and in vivo	Low processivity, time consuming, difficult equipment availability, need for complex analysis of results	Szulczewski et al. (2016), Ryabova et al. (2022), Neto et al. (2022), Kröger et al. (2022), Miskolci et al. (2022)	

Microfluidic methods for study of macrophages

Microfluidics has been an actively developing tool for conducting experiments in molecular and cellular biology over the past two decades. Microfluidics operate with small amounts of liquids and substances using hydrodynamic and small scale forces. Many conventional methods used in biology have been implemented in a microfluidic format. They are usually characterized by higher precision, faster rate, lower reagent consumption and higher level of automatization. The microfluidic format allows for precise control of the microenvironment, controlled application of stimuli, and real-time monitoring of processes that lead to more physiological conditions of in vitro studies. In practice, microfluidics is usually implemented using microfluidic chips—microdevices containing channels, chambers, active and passive hydrodynamic elements. Currently, a number of commercial microfluidic chip solutions are available on the market, but developing your own designs allows you to more accurately meet the task. The technical aspects of microfluidics remain beyond the scope of this review, while we focus on the advantages that microfluidics can offer in the study of macrophages.

Migration assays

Certain macrophage subtypes exhibit high motility associated with participation in immune responses. Conventional assays for cell motility include scratch test, transwell migration systems and more sophisticated impedance measurements, with specific pros and contras described elsewhere (Hulkower & Herber, 2011; Limame et al., 2012). Microfluidic cell migration assays are highly advantageous in creating controlled concentration gradients (Cooksey, Sip & Folch, 2009).

Deroy et al. (2022) used an approach based on a ‘fluid-walled’ microfluidics to evaluate the motility of murine macrophages towards chemoattractant component 5a (C5a). The approach uses extremely interesting fluid-shaping technology, which is described in detail in Soitu et al. (2020). Briefly, microfluidic circuits of the culture medium and immiscible fluorocarbon FC40 were printed in situ using a self-developed jet-printing device. Using this technology, Soitu et al. (2020) have studied murine bone-marrow derived macrophage migration behaviour in different regimes. First, a passive gradient of C5a was applied to cell clusters placed in different compartments of the fluid channel. Remarkably, only cells located at the frontline of a cluster migrated towards C5a, independently of the absolute position of a cluster within the concentration gradient. The minimum C5a concentration sufficient for chemotaxis was found to be lower than in more conventional settings. Next, macrophages with knocked-out C5a-receptor 1 react to chemoattractant significantly weaker than the wild-type cells. Finally, chemotaxis in active flowing gradients was investigated using an m-shaped circuit. Smaller concentrations of attractant could be reached under flow conditions. Also dynamically shifting gradient edge could be applied by varying flow rates. According to the data, macrophages can follow moving gradients less efficiently than static equivalents. This may indicate that macrophages sense the concentration gradient both in a spatial and temporal manner, which is particularly important for in vivo processes where chemokine gradients are likely evolving over time. Soitu et al. (2020) approached both the acquisition of new, previously unobserved results and the methodological and technical aspects of the study with great care. Special attention was given to the increase in scalability and throughput of the method. The ‘fluid-walled’ microfluidics technology has been considered for commercial applications in automated systems.

At the same time, the accessible 2D microfluidic devices can hardly imitate the native 3D microenvironments packed with extracellular matrix. Pérez-Rodríguez et al. (2022) overcome these limitations by creating a microfluidic chip for macrophage culture in hydrogel. The central channel of the chip contains cells in collagen hydrogel, separated from two side channels by multiple columns with 300 µm gaps. Side channels, one filled with culture medium and the other filled with bacterial fractions, create a stimuli gradient. Macrophage migration was studied using THP-1 monocyte cell line as precursors and various derivatives of pathogenic (M. tuberculosis H37Rv and S. typhimurium SV5015) and nonpathogenic (E.coli DH5α and M. smegmatis mc2155) bacteria as stimuli. The experiments revealed undirected (random) motility of macrophages in the absence of stimuli, at rates depending on collagen concentration in the hydrogel. Directed motility was encountered in response to specific derivatives of both pathogenic and non-pathogenic strains, with concentration-dependent directionality in a threshold mode demonstrated for certain stimuli. Different bacterial fractions induced more intense macrophage movement, leaving room for discussion of the specific molecular mechanisms underlying macrophage sensitivity.

One drawback of microfluidic migration assays with regard to macrophages is the lack of means for the distinction of molecular or functional subtypes. Considering the inherent sensitivity of macrophages to microenvironmental clues, including surface topography and hydrophilicity, the choice of materials for microfluidic chips should account for possible interference with macrophage adhesion capacities and phenotypes (Kosoff et al., 2018).

Single-cell study and droplet microfluidics

Primary human macrophages demonstrate a high heterogeneity in vivo, which often escapes the attention of conventional investigation methods. Droplet microfluidics provides a unique opportunity to study macrophages at a single-cell level (Li et al., 2023). The cells are encapsulated, individually or in small groups, in submicroliter droplets emulsified in a continuous phase of mineral oil. Different stimuli agents or drugs could be enclosed together with cells. The method is more suitable for suspension cultures of monocytic precursors than mature macrophages (Gencturk et al., 2022) as the adherent cell types may alter their functional profiles upon encapsulation (Jain, Moeller & Vogel, 2019). Tiemeijer et al. (2021) used thermo-reversible polyisocyanide hydrogel as a dispersed phase for human blood-derived macrophages in order to increase the viability and promote M2 polarization. A multicell culturing mode was shown to stimulate expression of M2 markers. Considerable heterogeneity of M2 pools was noted in the experiments, with certain M2-induced cells showing the increased CD80 and decreased CD206 expression, also secreted tumor necrosis factor-alpha (TNFα), specific to the M1 subpopulation.

Macrophage heterogeneity can manifest in differential response to various drugs and toxins. Ma et al. (2023) used droplet microfluidics combined to inductively coupled plasma mass spectrometry to study mercury accumulation and elimination in THP-1 macrophages cell line following thimerosal exposure. The single-cell measurements of Hg content revealed heterogeneous rates of uptake and clearance. Most cells showed low concentrations of uptaken mercury, while only <5% showed high concentrations. Upon withdrawal of exposure the cells continued to show high clearance rates for 10–15 h without complete elimination of the agent, probably causing long-term chronic toxicity. ROS and GSH levels were used to appreciate the toxicity effect. According to the results, at thimerosal concentrations under 50 ng/ml the redox balance is preserved despite a slight increase in ROS production, while 100–200 ng/ml of thimerosal cause oxidative stress continuing for 10–15 h since the exposure.

The study of cells at the single-cell level is of significant interest and marks a new frontier in cell biology research. Droplet microfluidics is a powerful tool that facilitates such investigations. However, it’s important to note that this technique demands highly precise flow control, presenting a considerable technical challenge.

Cell sorting

Another task, already solved for droplet microfluidics, is sorting (Xi et al., 2017; Huang et al., 2022). The precise stream manipulation allows selection and redirection of both droplets and individual cells, similarly to fluorescence-activated cell sorting (FACS) technology. By application of external forces, microfluidic cell sorting can be classified into ‘active’ and ‘passive’, the former exemplified by whole blood separation based on hydrodynamic properties only (Tripathi et al., 2015). In ‘active’ sorting, external forces of various nature are used as deflection stimuli. Zhang et al. (2020b) suggested to separate cells by their dielectrophoretic (DEP) spectrum. Displacements of RAW264.7 macrophages and MCF-7 cells under an alternating electric field of various frequencies were measured to determine the optimal separation conditions. Using a microfluidic chip with DEP deflection force, Zhang et al. (2020b) showed above 99% separation effectivity of RAW264.7 macrophages from MCF-7 cells.

Alternatively, the streamed cells can be deflected by optical forces similar to those used in ‘optical tweezers’. Perroud et al. (2008) used a 1,024 nm infrared laser with 9.6 W power for fluorescent-dependent sorting of RAW264.7 macrophages cell line. The cells were selectively deflected by a pulsed laser beam from a hydrodynamically focused cell sequence. In verification experiments, red and green fluorescently labeled cells were sorted with 97% accuracy at a rate of 14–22 cells per second. The method afforded efficient sorting of macrophages infected with F. tularensis subsp. novicida labeled with amine-reactive Alexa-488 succinimidyl ester. Cell viability, activation and functionality tests were carried out to exclude an adverse effect of infrared laser on murine macrophages.

Magnetic field can be used as a deflecting force for cells with inherent or extrinsic magnetic moments (Myklatun et al., 2017). The accuracy of sorting will depend on the magnetic field intensity gradient and interaction time values. To significantly enhance separate efficiency of macrophages with magnetite nanoparticles ingested by phagocytosis, Myklatun et al. (2017) introduced a side flow of dense ferromagnetic fluid in close vicinity to the cell sorting channel. The sorting efficiency varied from 21% to 90% for flow rates of 2 μm/min to 0.5 μm/min, respectively. In contrast, the permanent magnet and absence of an external magnetic field showed only 6% and 2% correspondingly at a flow rate of 1 ul/min. Similar sorting of magnetotactic bacteria M. gryphiswaldense in a scaled-down microfluidic chip used visual observation for the tuning of sorting parameters. Although this method showed rather low throughput, it could be further improved and could be applicable for non-contamination sorting of magnetic cells.

Co-culture chips and organ-on-a-chip platforms

Studies of interactions of macrophages with other cell types in vitro are often challenging at the level of individual factors due to complex side effects that arise in mixed cultures. Microfluidic devices enable spatial separation of cell types in a culture while ensuring their communication via culture medium. The principle was used by Li et al. (2022a) to study paracrine interactions between fibroblasts and macrophages during in vitro wound healing. The restoration of a fibroblast layer was simulated in a co-culture chip containing RAW264.7 macrophage cell line. The design excluded direct contacts between fibroblasts and macrophages apart from diffusion of secreted factors via narrow channels. M2 polarized macrophages were found to significantly activate fibroblasts, promote their migration to the wound site and stimulate F-actin and α-smooth muscle actin expression, whereas M1 cells had no such effect.

One of the most exciting and prospective types of microfluidic chips is known as organ-on-a-chips (OoC). The organ-on-a-chip microfluidic devices mimic the basic architecture or/and functions of an organ to maximize the similarity with natural microenvironments (Leung et al., 2022). Such microfluidic models provide the next step in approaching in vitro experiments to physiological in vivo conditions. The design can be particularly useful in drug discovery, pathology biomodeling, and fundamental understanding of physiological processes in organisms (Ingber, 2022). A study on the human liver macrophage polarization under bacterial infection and persistence used a liver-on-a-chip microfluidic scaffold populated with hepatocytes, endothelial cells and macrophages (Siwczak et al., 2022). The presence of macrophages was found to significantly mitigate the uptake of S. aureus by hepatocytes and endothelial cells, so macrophages protect the organs from the infections similar to the in vivo. At the same time, M2 polarized macrophages showed higher levels of ingested bacteria at all stages of the modeled infectious process, whereas a switch to small-colony bacterial phenotypes inside macrophage cells required the presence of hepatocytes and endothelial cells in co-culture. Therefore, the mechanism of persistence of bacteria and escape of elimination by immune cells was highlighted.

Landau et al. (2024) used human primitive macrophages differentiated from pluripotent stem cells in a commercially available BioWire and iFlow platforms for heart-on-a-chip modelingto demonstrate a strongly positive role of macrophages in the heart tissue microvascularization and perfusion. Lung-on-a-chip models with incorporated murine macrophages were used to study inflammatory processes of viral or bacterial nature in the lungs (Thacker et al., 2021, 2020). Intestine-on-a-chip models were used to study the role of macrophages in human gut inflammation (Gijzen et al., 2020; Beaurivage et al., 2020).

The growing interest in macrophages as tools and targets for cancer therapy (Mantovani et al., 2022) underscores the relevance of tumor-on-a-chip microfluidic devices populated with cancer cells, macrophages and a modeled microvascular network (Liu et al., 2021). Bi et al. (2020) performed a tumor-on-a-chip comparative study with cell lines derived from colorectal cancer and a less aggressive pancreatic ductal adenocarcinoma. The addition of M1 macrophages differentiated from THP-1 cell line triggered a strong antitumor effect in both tumor types. By contrast, M2 polarized macrophages stimulated the malignant behaviors of cancer cell lines in both systems without affecting angiogenesis. Proteomic analysis of the flow-through medium associated soluble factors CXCL9, CXCL10, CXCL11 with the anti-tumor effect of M1 cells and MMP7, ANGPT2, CCL-3, CSF-1 with the opposite effect of M2 cells (Bi et al., 2020). Manoharan et al. (2024) assessed the involvement of unpolarized macrophages derived from THP-1 cell line and T cells in cancer progression: breast cancer cells were found to promote the pro-tumor M2 polarization in a tumor-on-a-chip systems, while T cells supported the anti-tumor M1 phenotypes in co-cultured macrophages. It can be concluded that the interaction of the tumor with the immune system plays a critical role in tumor progression. That is why it is very attractive to turn macrophages into the means of antitumor drug delivery. Wang et al. (2020) used tumor-on-a-chip to address the prospects of macrophages as a vehicle in drug delivery. Murine macrophages RAW264.7 cell line loaded with carrier nanoparticles showed the ability to directionally migrate and infiltrate a densely packed tumor spheroid embedded in collagen gel (Wang et al., 2020).

Microfluidic technologies offer extensive potential for modeling biological processes in vitro. They not only open new avenues for studying macrophages and other cell types but also address ethical concerns related to research involving laboratory animals. Additionally, organ-on-a-chip and tumor-on-a-chip models inherently utilize cells from the desired origin (human or animal). However, this technology still requires substantial validation through studies comparing results with whole organs and organisms.

Microfluidics-based methods used in macrophage studies are summarized in Table 2.

Table 2 Comparative characterization of the m icrofluidic methods used in macrophage studies.

Microfluidic technology	Opportunities	Strengths	Weakness	References	
Migration assay	Microfluidic chips for 2D and 3D cells migration study	Precise control of chemical, thermal, and mechanical gradients in 2D or 3D	Non-accounting for cell heterogeneity; difficult to compare results due to individual chip designs	Pérez-Rodríguez et al. (2022), Deroy et al. (2022)	
Droplet microfluidics	Encapsulation of cells in emulsified submicroliter droplets	Provides high-throughput system for single-cell analysis	Needs precise flow control; uses two-phase emulsion, which can alter the cell functionality; low cell viability and load density	Tiemeijer et al. (2021), Ma et al. (2023)	
Cell sorting	Separation of cells by size, fluorescence, magnetic moment, etc.	High-throughput
tunable method, may allow cell sorting without labeling	Individual experiment design for different cells	Perroud et al. (2008), Myklatun et al. (2017), Zhang et al. (2020b)	
Organ-on-a-chip	Microfluidic chips mimicking organ architecture and composition	High-tech in vitro model useful in fundamental research and pre-clinical screening	Still needs verification in relation to in vivo conditions	Thacker et al. (2020, 2021), Gijzen et al. (2020), Beaurivage et al. (2020), Siwczak et al. (2022), Lagowala et al. (2024), Landau et al. (2024)	
Tumor-on-a-chip	Microfluidic chips mimicking tumor architecture and composition	High-tech in vitro model useful in fundamental research and pre-clinical screening	Still needs verification in relation to in vivo conditions	Cui et al. (2018, 2020), Bi et al. (2020), Chernyavska et al. (2022), Manoharan et al. (2024)	

Methods for study of macrophages mechanical properties

Macrophages are motile cells highly sensitive to mechanical clues, with mechanical properties adaptable to dynamic microenvironments. Several advanced techniques can be applied to study mechanical properties of individual macrophage cells, including atomic force microscopy, single cell and cell-to-cell force spectroscopy and ‘optical tweezers’ (Fig. 2).

Figure 2 Biophysical methods for studying cell nanomechanics.

(A) AFM nanoindentation provides data on cell stiffness by analyzing tip-to-object approach curves; (B) single-molecule force spectroscopy measures interaction forces at the single-molecule level; (C) cell-to-cell force spectroscopy examines complex interactions between entire cells; (D) optical tweezers offer a unique opportunity to study the dynamic rheological properties of cells through interactions with optically trapped particles.

Scanning probe microscopy for cell stiffness measurements

Although optical microscopy remains the gold standard in cell biology, it cannot provide direct information on elasticity or force of adhesion in cell layers, even less so for isolated cells. Atomic force microscopy (AFM), which gives a combined output on both cell morphology and mechanical properties, has been particularly useful in macrophage studies (Rotsch et al., 1997). With the advent of new techniques and analytical methods, it has become possible to obtain new unique information about both morphology and cell mechanics (Bitler, Dover & Shai, 2012; Tian et al., 2019). Souza et al. (2014) used AFM nanoindentation technique to measure elastic moduli of murine macrophages cell line J774 on modified substrates before and after cytochalasin D treatment. Macrophages cultured on uncoated glass had lower Young’s moduli than those cultured on fibronectin, which indicated a modulatory role of extracellular matrix in cell stiffness. Furthermore, treatment with cytochalasin D, a potent actin depolymerization agent, caused cell softening, which confirmed the association of cytoskeletal remodeling with mechanical properties. Labernadie et al. (2010) used AFM correlative fluorescence microscopy to study podosomes of macrophages derived from blood monocytes—unique cellular structures involved in transient interactions with extracellular matrix (Fig. 3). The height and average stiffness of podosomes (578 ± 209 nm and 43.8 ± 9.3 kPa, respectively) in human macrophages cultured on microstructured substrates composed of different extracellular matrix proteins revealed no correlation with the protein identity, albeit fibronectin promoted higher rates of podosome formation compared with other substrates. Furthermore, two podosome stiffness variations were observed at frequencies of ~0.14 and ~0.031 Hz, probably depending on F-actin integrity, actin treadmilling rates and myosin II activity. Further studies on macrophages adhered to a thin Formvar sheet allowed to estimate the force generated by a single podosome on the substrate surface. The magnitude of the force was assessed by the height of the protrusions formed by podosomes on the reverse side of the sheet, measured using AFM. Force occurs to correlated with the substrate stiffness. Labernadie et al. (2014) called this variation of the technique protrusion force microscopy.

Figure 3 Using atomic force microscopy for study the macrophage podosomes.

(A) AFM deflection image of human macrophage on fibrinogen micropatterned substrate; (B) AFM height and deflection images of podosomes formation on fibrinogen coated spot; (C) height cross-section of macrophage podosomes, formed on fibrinogen coated spot. Modified from Labernadie et al. (2010) exclusive PNAS License to Publish.

Returning to whole cell properties, were performed by Suleimanov et al. (2024) studied oxidative activity and cell stiffness in human macrophages derived from blood monocytes with regard to phenotype. Polarization and activation of macrophages were accompanied by changes in cell morphology and stiffness associated with F-actin remodeling. Activation of M1 and M2 polarized macrophages with PMA promoted a significant increase in their Young’s moduli, while stiffness of non-polarized M0 macrophages remained unchanged. The power-law rheology model revealed a transition of cells to a more rigid state with enhanced cytoskeletal prestress characteristic of spreading. Another scanning microscopy method termed scanning ion-conductance microscopy (SICM) demonstrated an increase in THP-1 macrophage cell line stiffness after exposure to low-density lipoprotein extracted from blood samples (Fig. 4) (Kiseleva et al., 2024).

Figure 4 Topography and Young’s modulus mapping obtained by scanning ion-conductance microscopy on THP-1 cell line macrophages before and after exposure with low-density lipoprotein.

Modified from Kiseleva et al. (2024), CC-BY 4.0.

Force spectroscopy for cell interaction measurements

Force spectroscopy is another scanning probe microscopy technique which uses the probe, modified with a cell, bacteria, or molecule to obtain retractive curves to determine the force of interaction. Forces like adhesion play a crucial role in macrophage activity, as these cells are part of the immune system. Force spectroscopy is one of the few methods capable of directly measuring the magnitude of such forces, owing to the cantilever’s exceptional sensitivity to applied force. Targosz et al. (2006) used force spectroscopy to assess the interaction of murine macrophage surface receptors with single molecules of bacterial wall components lipopolysaccharide and exopolysaccharide. The measured adhesion force of ~102 pN (piconewtons) was found to correlate with macrophage activation status (Targosz et al., 2006). Similar dynamics of adhesion forces between lipopolysaccharide and RAW264.7 macrophages cell line were demonstrated by Pi et al. (2016) The average adhesion force decreased after treatment of the cells with dexamethasone and quercetin (Pi et al., 2016). El-Kirat-Chatel & Dufrêne (2016) studied interaction forces between a fungal pathogen C. albicans and murine macrophage cell line J774A.1. The force spectroscopy measurements revealed two types of bonds formed by yeast particles: short-range molecular bonds and long-range tethers up to 100 µm in length. The overall force of adhesion could reach 3,000 pN and tended to increase with the time of contact (El-Kirat-Chatel & Dufrêne, 2016). Li et al. (2013) measured adhesion forces between the modified tip and Fc gamma receptors at the surface of RAW264.7 macrophages and assessed the nanoscale distribution of the receptors at cell surface by force spectroscopy mapping.

Optical tweezers cell-cell interaction measurements

Alternative means to measure mechanical effects at the single-cell level are provided by ‘optical tweezer’ instruments that entrap and manipulate micron-size particles with focused laser beam (Favre-Bulle & Scott, 2022). Unlike force spectroscopy, optical tweezers do not have a clear requirement for cell adhesion to the substrate. This allows one to avoid the influence of the substrate and measure the mechanics and rheology of monocyte precursors (Fore et al., 2011). The technique enabled clamping of single macrophage cells between 500 nm beads functionalized with concanavalin A for stiffness and rheology measurements. The experiments found M2 human blood-derived macrophages stiffer and more viscous compared to M1 (Evers et al., 2022). Several studies used optical tweezers technique to determine the force of adhesion between macrophages (J774 and THP-1 cell lines) and LPS-functionalized beads (Wei et al., 2007; Su & Hsu, 2010; Byvalov, Kononenko & Konyshev, 2018). The measured adhesion forces were slightly lower than corresponding values measured by force spectroscopy (Targosz et al., 2006), the discrepancy probably reflecting different sources of LPS and macrophages used in the experiments.

As with all methods for investigating mechanical properties, it is important to recognize that the results are highly dependent on the experimental design and techniques employed (Wu et al., 2018). Some approaches rely on mathematical models to extract characteristics from the raw data. While these studies are invaluable, the results should be interpreted as relative values, emphasizing the need for carefully planned control experiments.

So, we should keep this point in mind and not accept the obtained values as absolute, but try to conduct well-organized comparative experiments.

Conclusions

Macrophages are complex but interesting cells, striking in the diversity of their phenotype and population heterogeneity even within a single organ. Specialists working with macrophages are like doctors working with a newly discovered disease: they constantly need to expand the range of research methods. This review contains several examples of a new point of view on such a classical biological object as a macrophage. For example, FLIM provides not only spatial information like classic fluorescence microscopy, but also temporal resolution, which often escapes attention. The mechanical properties of cells also often do not take into account in cell phenotyping while they have a primary influence on cell motility and functions. The single cell level of analysis in contradiction to population level is of particular importance for such heterogeneity objects as macrophages. In this Review a number of biophysical analytical methods are represented, which fill described gaps and provide an all-round view on macrophages in combination with classical cell biology methods. The use of new approaches at the intersection of biological and physical sciences, their reduction in cost and distribution will expand our understanding of the physiology of macrophages, which began with the description of the role of phagocytosis in immunity by Mechnikov in the late 19th century.

Additional Information and Declarations

Competing Interests

The authors declare that they have no competing interests.

Author Contributions

Polina Vishnyakova conceived and designed the experiments, performed the experiments, analyzed the data, prepared figures and/or tables, authored or reviewed drafts of the article, and approved the final draft.

Andrey Elchaninov conceived and designed the experiments, performed the experiments, analyzed the data, prepared figures and/or tables, authored or reviewed drafts of the article, and approved the final draft.

Timur Fatkhudinov conceived and designed the experiments, analyzed the data, authored or reviewed drafts of the article, and approved the final draft.

Dmitry Kolesov conceived and designed the experiments, performed the experiments, analyzed the data, prepared figures and/or tables, authored or reviewed drafts of the article, and approved the final draft.

Data Availability

The following information was supplied regarding data availability:

This is a literature review.

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
