# Peer review of "Unravelling approaches to study macrophages: from classical to novel biophysical methodologies"

_PeerJ, doi:10.7717/peerj.19039_

## Round 0.1 · original submission · Major Revisions

This manuscript requires a thorough revision. Please address comments of all reviewers and provide responses in a point wise manner.

Reviewer 1 ·

Basic reporting

Some English editing and putting the references are needed.
Structure and figures are OK

Experimental design

No comment

Validity of the findings

This will be benefits to the readers.

Additional comments

Vishnyakova et al. reported interesting knowledge on macrophages focusing on the method to study. The review topic is unique and helpful for the readers in the field of study. I only have several minor comments on the manuscript.
1. The English language needs some corrections.
2. Some sentences need to be confirmed. “Macrophages stand out from other cells due to their presence in all types of the organism…”. Is it all organisms or all sites in the human body?
3. Please put appropriate references in the sentences mentioning all macrophages (line 58-62).
4. In the topic “Origin and role of macrophages”, how about adding the cell lines or cells for studying macrophages to complete the manuscript. The author already interestingly mentioned RAW, J774A.1, THP-1, etc. in the review.
5. The M1/M2 paradigm is incomplete. There are overlapping of M1/M2 subtypes, for example, Strizova Z. M1/M2 macrophages and their overlaps - myth or reality? Clin Sci (Lond). 2023 Aug 14;137(15):1067-1093, that should be mentioned. Currently, it seems that this part is not going along with other parts that are focusing on the method to study macrophages. Some sentences to link M1 M2 and new methods for study should be added. Of note, I agree with putting M1 M2 in this part because the authors mentioned it in nearly all subsections; however, a linkage is needed for the flow of reading to let the reader know why M1 M2 is here.
6. Survey methodology should not be a separate topic, I recommend putting this part as an opening of the section “methods of studying macrophages”.
7. Line 147 and 148, what is another population of kidney macrophage? The sentence seems to be incomplete.
8. Line 164, there are differences between transcriptome and RNA seq (you should spell this out in the 1st use). The declaration of terminology might be good.
9. Please put appropriate references in several sites in the section mass spectrometry and single-cell mass spec (line 211-269). A lot of sentences do not have ref. at the end of the sentences.
10. Line 292-293, the definition of microfluidic methods will help the reader understand the review. For example, migration assay and cell sorting can be done simply by conventional in vitro experiments or flow without the use of microfluidic methods mentioned here. Also, please spell out in the 1st use of the abbreviation, here, are “FACS” and “ECM”.
11. Line 484, the example of images from some of these methods will enhance the interest in this review.
12. Line 513, “performed by S.K.Suleimanov and Y.M.Efremov and coauthors”, this is not a standard method to mention a previous work. How about simply “Suleimanov et al.”?

Reviewer 2 ·

Basic reporting

The review presents a collection of different methods commonly used in biology that can also be used to study macrophages. The authors did not explain which techniques can be combined to obtain additional information on macrophage phenotypes, functions, etc. The authors did not mention the differences in macrophage populations/types. The title doesn't fit to the content. The description of (partly new) biophysical methods is only a small part of the manuscript.

Experimental design

Google and PubMed searches were performed; this is not a systematic review, although some inclusion and exclusion criteria were provided. The studies mentioned were on cell lines that do not have the full potential of normal cells.

Validity of the findings

The advantage of using and/or combining different technologies is not clear.

Additional comments

The motivation for the study and the information obtained are not clear. Figure 2 is only focused on the methodology in general with no specific link to macrophages. Figure 1 provides too little detail.

·

Basic reporting

The manuscript aims to describe the main methodologies and techniques currently used for studying the biology and the functionality of the main innate immune cells, i.e., macrophages. Overall, the manuscript is well conceived/designed, and the English is professional.
However, even if the intent of the authors to provide the description of methods is good and useful for who wants to approach the investigation of these cells, the review should be improved to fully reach this goal as explain below.

Experimental design

First, it is extremely important to distinguish murine and human macrophages, they are obviously very similar for functions and biology, but they can be also diverse in terms of different sensitivity to the same stimulus, specific markers, origin and so on. So, the entire description of M1 and M2 macrophages should be re-written by highlighting the difference between murine and human macrophages. Some markers are specific for M1 or M2 murine macrophages but not for human macrophages and vice versa. Moreover, the discrimination between M1 and M2 based on the two ways of arginine metabolism is still debated for human macrophages in vitro while it is well known for murine macrophages. So, throughout the manuscript, the authors should always discriminate between the macrophages from the two species and specify which markers are used for studying murine macrophages and which for human ones and which for both.
In line 126, the authors declare that the most papers evaluated are based on works performed on cell lines. This penalized the value of the manuscript especially considering that in literature a lot of investigations related to macrophages have been performed on bone-marrow derived macrophages for mice or monocyte-derived macrophages for humans, in both cases on primary cells.
So, in line with what I comment above, I suggest the authors to deeply check in the literature and to revise the manuscript by specifying which markers or methodologies are more used for murine or human macrophages and what has been investigated by using cell line or primary cells.

In line 152, please specify which cytokines the authors are referring to.

Regarding the tables, I suggest improving them by showing SWOT analysis of the different methods (strengths, weakness, opportunities, threats and references)

Regarding the figure 1, please improve it by highlighting markers for human and murine macrophages, and for both figures, please add a more descriptive legend.

Overall, for the sections on new microfluidic methods and methods for cell mechanical properties (both very important and interesting), I suggest the authors avoid the description of the works cited and their main results, but they should only describe and explain the methods considering for what they are used, and deeply discussing the swot analysis also summarized in the tables (see comment above).

As last, considering that the first part of the review is dedicated to classical methodologies (so it is a relevant section), I suggest the authors change the title. For example, something like “unravelling the approaches to study macrophages: from classical to novel biophysic methodologies”. This is just an idea, the authors are free to change as they prefer.

Validity of the findings

Please, see the comments in section 1 and 2.
The manuscript could be very valid for content and meaning but it should be deeply improved.

Additional comments

No further comments.

---

## Round 0.2 · accepted · Accept

Authors have addressed all of the reviewers' comments and manuscript is ready for the publication.

Reviewer 1 ·

Basic reporting

clear

Experimental design

clear

Validity of the findings

clear

Additional comments

I have no further comments.

Reviewer 2 ·

Basic reporting

The revision addressed the comments in appropriate form.

Experimental design

The revision addressed the comments in appropriate form.

Validity of the findings

The revision addressed the comments in appropriate form.